# Effects of Phosphate-Enriched Nutrient in the Polyculture of Nile Tilapia and Freshwater Prawn in an Aquaponic System

Soko Nuswantoro [1,2,†] [ID], Tzu-Yuan Sung [3,†], Meki Kurniawan [3], Tsung-Meng Wu [4], Bonien Chen [1,*] and Ming-Chang Hong [3,*] [ID]

1   Institute of Aquatic Science and Technology, College of Hydrosphere, National Kaohsiung University of Science and Technology, Kaohsiung 811213, Taiwan
2   Study Program of Aquaculture, Faculty of Fisheries and Marine Science, Brawijaya University, Malang 65145, Indonesia
3   Department and Graduate Institute of Aquaculture, College of Hydrosphere, National Kaohsiung University of Science and Technology, Kaohsiung 811213, Taiwan
4   Department of Aquaculture, National Pingtung University of Science and Technology, Pingtung 91201, Taiwan
*   Correspondence: cbnabalone@nkust.edu.tw (B.C.); junkrough.hmc@nkust.edu.tw (M.-C.H.); Tel.: +886-910238566 (B.C.); +886-7-3617141 (ext. 23721) (M.-C.H.)
†   These authors contributed equally to this work.

**Abstract:** Aquaponic systems are made up of hydroponic beds and recycled aquaculture systems. The significant elements that determine how effectively an aquaponic system operates are the nitrogen cycle (nitrification) and the phosphorus cycle (phosphate). Because some research indicates that aquaponics systems are primarily deficient in phosphorus, phosphate-enriched nutrients were added to raise the phosphorus levels. During an eight-week experimental period, the effects of water quality parameters and microbiology, animal and plant growth performance, chlorophyll compounds in lettuce, and the bacterial community were analyzed. Phosphate concentration ($1.604 \pm 1.933$ mg L$^{-1}$) and ammonia-oxidizing bacteria (AOB) ($1.19 \times 10^2 \pm 1.30$ CFU mL$^{-1}$) give significant positive reactions to the added nutrients. However, the prawn survival rate ($17.00 \pm 0.63\%$) showed a significantly negative response to nutrition modification containing phosphate, and the percentage of bacterial pathogens became more dominant (pathogen 40.51%; N_bacteria 35.05%; probiotics 24.44%). This study shows that adding phosphate increases phosphorus levels in an aquaponics system and changes the microbial community and species growth performance.

**Keywords:** aquaponic system; phosphate deficiency; phosphorus level; microbial community





## 1. Introduction

Aquaponics is an approach for merging aquaculture and hydroponics in an environment that grows crops in recycled aquaculture water [1]. Hydroponics is the practice of growing plants in water without soil. The two are combined in an enclosed, recirculating system called aquaponics. The organic wastes accumulated in the water are filtered and removed with a typical recirculating aquaculture system, keeping the water clean for the fish. However, an aquaponics system passes the nutrient-rich wastewater through a plant-filled inert substrate. Here, microorganisms break down the waste from the fish, plants absorb the resulting nutrients, and finally, the water is cleaned and returned to the fish tanks [2]. By cleaning the water to dispose of harmful waste and reusing water, recirculating systems are intended to rear mass amounts of fish in relatively small amounts [3].

In soils, natural water streams, and municipal sewage, nitrification is vital in cycling nutrients [4]. Nitrification results in the oxidation of ammonia to nitrate in agricultural land fertilized with ammonia-based fertilizers [5]. Through nitrifying microbes, $NH_4^+$ is easily converted to $NO_3$ in oxic conditions, such as deep-cleaning soils. Due to the ease with which plants can absorb nitrate, the process is crucial for soil health [6].

In general, in addition to the nitrogen cycles with nitrification, phosphorus cycles in aquaponics settings cannot be separated. Phosphorus, an essential mineral, frequently determines biological mechanisms, including crop growth, in soils [7]. As it is a necessary component for supporting all living and agricultural production on our earth, phosphorus (P) is a critical component. All crops require three macronutrients (N, K, and P), and it is one of them [8]. Crop production is slowed by phosphorus deficiency, while too much phosphorus might react negatively with other nutrients [9].

Previous investigations have discovered a deficiency in total P in their aquaponics systems [10,11]. This could be because P precipitated as solids, separated in purification chambers, and rendered inaccessible. If the system water has much calcium, orthophosphate might precipitate once at neutral pH to generate calcium phosphate or dicalcium phosphate [12,13]. When calcium phosphate ($Ca_3(PO_4)_2$) is in a solid form, it will not be possible for crops to utilize it, and it can be removed from the system with a purification chamber [14]. When crops cannot utilize P in solid form, they will grow slower because of a lack of P uptake, primarily in higher nutritional crops such as fruits or vegetables [15].

The purpose of this study was to evaluate the effect of the addition of phosphate-enriched nutrients (MN) and without (NN) in an aquaponic system, using lettuce (*Lactuca sativa* L.) and tilapia (*Oreochromis niloticus*) maintained with freshwater shrimp (*Macrobrachium rosenbergii*), on water quality and microbiology, aquaponics performance (tilapia, giant freshwater prawn, and lettuce), chlorophyll compounds in lettuce, and the bacterial community using a metagenomics technique.

## 2. Materials and Methods

### 2.1. Aquaponics Design

The aquaponics system comprised two 500 L fish tanks, a 1 × 4 × 0.5-m hydroponic tank with a water volume of 1200 L and hydroton (8–12 mm) at a minimum depth of 30 cm, a 2 × 1 × 0.5-m sump tank with a water volume of 400 L, and two filtration tanks (a solid separator and biofilter) with a water volume of 200 L each. Water from the fish tanks flows through a solid separator and biofilter (equipped with a K1 filter, an aerator, and live bacteria) before reaching the media bed where plants are grown. The water then flows to the sump tank and is recirculated back to the fish tanks. The system uses gravity to move water and a bell siphon to automatically flood and drain the bed media. A water pump (SP500 AH, 1/2 HP, 110/220 V) with a flow rate of 600 L per hour and a ball valve regulates the flow rate from the sump tank back to the fish tanks (Figure 1).

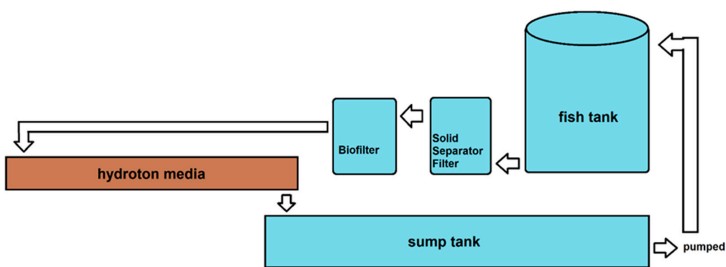

**Figure 1.** Aquaponic water flows experimental design.

### 2.2. Phosphorus Solution Preparation

A phosphorus solution was made by mixing urea (30 g), calcium nitrate (236 g), monopotassium phosphate (90 g), monoammonium phosphate (37 g), magnesium sulfate (246 g), chimeric iron (10 g), boric acid (3 g), manganese chloride (1 g), zinc sulfate (0.09 g), copper sulfate (0.04 g), and sodium molybdate (0.01 g) in 1 L of distilled water to make a 1000× phosphorus solution [16]. The solution was only given at the beginning of the experimental period when the lettuce was planted to raise the phosphate level.

### 2.3. Experimental Setup

The study was conducted using six aquaponic setups, with three serving as the control group with unmodified nutrition (NN) and three as the treatment group with modified nutrition (MN). In each setup, 100 tilapias ($100 \pm 2$ g) were placed in both of the two fish tanks, and 30 prawns ($20.0 \pm 0.3$ g) in the sump tank. Once the system is running smoothly and stable for two weeks, 50 lettuce plants (*Lactuca sativa* L-Known-You Seed) can be transplanted into the grow bed of the aquaponic system, one week after they have been seeded in seedling trays with compost media. An eight-week experiment was conducted from January to March 2020 at the Aquaculture Department of National Kaohsiung University Science and Technology in Kaohsiung, Taiwan, using open-space outdoor laboratories. During the experiment, tilapia and giant freshwater prawns were fed a 32% crude protein commercial diet (Quanxing 1.6 mm, Grobest, Pingtung County, Taiwan) twice daily, with the amount given proportionate to 3% of their body weight.

### 2.4. Water Quality Monitoring

Temperature, pH, and dissolved oxygen were measured twice daily (8:00 am and 5:00 pm) using HI98194 (HANNA Instruments, Smithfield, RI). Ammonia, nitrite, nitrate, and phosphate levels were measured weekly using HI83300 (HANNA Instruments, Smithfield, RI) [17]. The measurements were taken in each of the water columns (fish tank, filter tank, sump tank) and an average was calculated for each water quality parameter.

### 2.5. Microbiology

At the end of the experiment, 50 mL of sediment was collected from the aquaponic system's sediment bed for microbiological assessment. The samples were homogenized and diluted in a serially ($10^{-1}$, $10^{-2}$, $10^{-3}$) sterile solution (0 ppt) before being plated on selective media for ammonia-oxidizing [18] and nitrifying-oxidizing bacteria [19]. The samples were incubated at 35 °C, and colony-forming units (CFU) were counted after 72 h. In addition, hydrotone granules were collected, crushed, and preserved at $-5$ °C for microbial species identification using Next Generation Sequencing (NGS) in Tools Biotechnology (Taipei, Taiwan). Descriptive analysis was conducted to determine the variation of the bacterial population using the OTU bar chart. Three replicates of samples were collected at randomly selected positions.

### 2.6. Tilapia, Giant Freshwater Prawn, and Lettuce Growth Performance

The performance of tilapia and giant prawns was assessed weekly by taking samples of 10 fish/prawn from each tank. The weight of the fish/prawn sample was measured at the beginning and end of the experiment. Survival (percentage) and final mean weight (g) were measured at the end of the rearing period. At the end of the experiment, the lettuce was weighed. Overall biomass and productivity of fish, prawns, and lettuce were also assessed to calculate total productivity.

### 2.7. Chlorophyll-a Content of Lettuce

Grinding–settling (GS) methods can be used to extract chlorophyll [20], which involve immersing the sample in 95% aqueous ethanol and then measuring the amount of chlorophyll present using an optical density spectrophotometer (OD). The spectrophotometer measures the concentration of both chlorophyll a and b by measuring the optical density at 645 nm and 663 nm [21], respectively. This information is then used to calculate the total concentration of chlorophyll (chlorophyll a and chlorophyll b) in the sample. Three replicates of samples were collected at randomly selected positions.

### 2.8. Data Analysis

Statistics were completed using SPSS Software. Before statistical treatments, Shapiro–Wilk and Levene tests were used to determine data normality and homogeneity, respectively.

Before analysis, microbiological data were Log10 transformed to normalize the data [22]. The Student's t-test was then used to compare variables, considered significance at $p < 0.05$.

## 3. Results

### 3.1. Water Quality and Microbiology

During the experiment, the average temperature was $23.78 \pm 3.27$ °C (NN) and $23.08 \pm 3.12$ °C (MN), the dissolved oxygen was $7.37 \pm 0.82$ mg L$^{-1}$ (NN) and $7.36 \pm 1.04$ mg L$^{-1}$ (MN), and the pH was $7.59 \pm 0.30$ (NN) and $7.52 \pm 0.29$ (MN). Averages for ammonia, nitrite, nitrate, phosphate, and microbiological results are shown in Table 1. The weekly values for ammonia, nitrite, nitrate, phosphate, and microbiological results are shown in Figure 2.

**Table 1.** Chemical and microbial loads in an integrated aquaponic culture system.

| Parameter | NN | MN | *p*-Value (*t*-Test) |
|---|---|---|---|
| Ammonia (NH$_3$) (mg L$^{-1}$) | $0.30 \pm 0.21$ | $0.42 \pm 0.23$ | 0.074 |
| Nitrite (NO$_2{}^-$) (mg L$^{-1}$) | $0.18 \pm 0.21$ | $0.21 \pm 0.27$ | 0.654 |
| Nitrate (NO$_3{}^-$) (mg L$^{-1}$) | $0.96 \pm 1.46$ | $1.05 \pm 2.00$ | 0.857 |
| Phosphate (PO$_4{}^-$) (mg L$^{-1}$) | $0.72 \pm 0.58$ | $1.60 \pm 1.93$ | 0.040 * |
| Total bacteria | | | |
| AOB (CFU mL$^{-1}$) | $0.09 \times 10^2 \pm 0.06$ | $1.19 \times 10^2 \pm 1.30$ | 0.000 * |
| NOB (CFU mL$^{-1}$) | $2.26 \times 10^4 \pm 3.31$ | $3.42 \times 10^4 \pm 2.83$ | 0.198 |

Data presented as mean $\pm$ standard deviation (n = 3). CFU: Colony-forming unit. Asterisks (*) indicate statistical differences ($p < 0.05$).

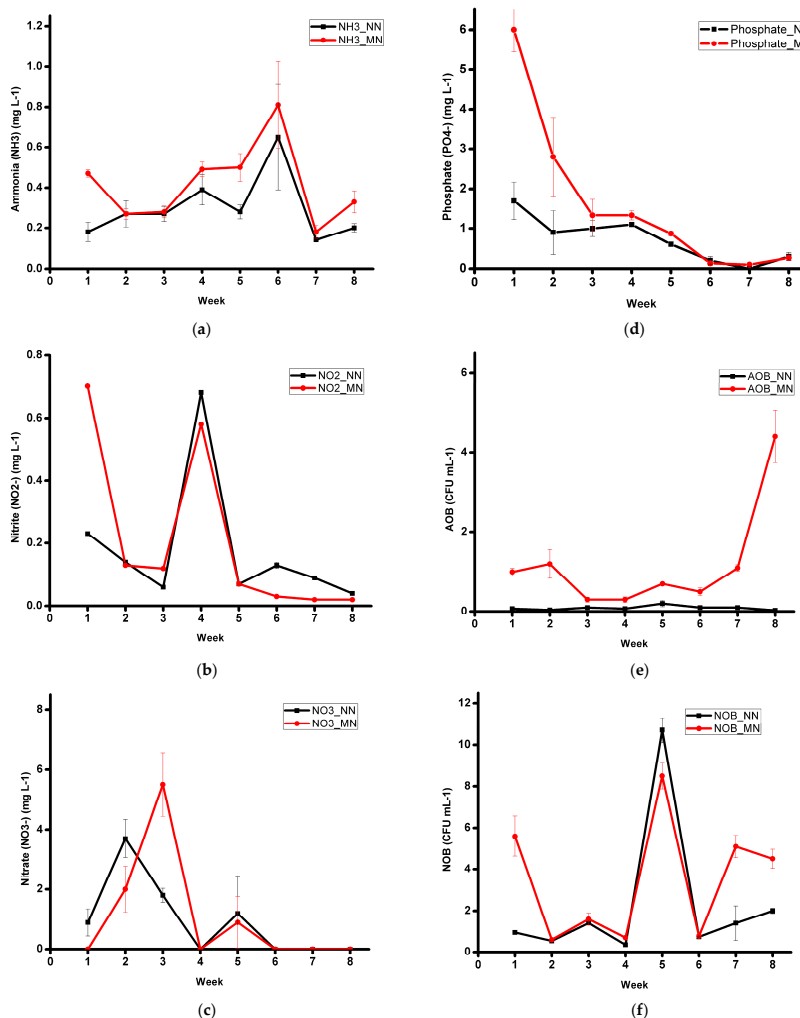

**Figure 2.** Weekly values for (**a**) ammonia, (**b**) nitrite, (**c**) nitrate, (**d**) phosphate, (**e**) AOB, (**f**) NOB (n = 3).

### 3.2. Tilapia, Giant Freshwater Prawn, and Lettuce Growth Performance

After eight weeks of rearing, the integration of prawn, tilapia, and lettuce with and without nutritional addition for the soilless culture influences prawn, tilapia, and lettuce performance, except for the survival of tilapia (Table 2). Prawn performance for the treatment with additional nutrition is worse than without nutrition. Prawns lose mass during culture in both treatments. However, the most significant loss after treatment is with additional nutrition—the lowest prawn survival was in the aquaponic setting using additional nutrition with only $17.00 \pm 0.63\%$ surviving. At the end of the experiment, $1.90 \pm 0.37$ kg of lettuce per experimental unit was harvested for the treatment without additional nutrition, which is more significant than the $1.65 \pm 0.42$ kg with additional nutrition.

**Table 2.** Performance in an integrated giant freshwater prawn (*Macrobranchium rosenbergii*), tilapia (*Oreochromis niloticus*), and lettuce (*Lactuca sativa* L) system (n = 3).

| Parameter | NN | MN | *p*-Value |
|---|---|---|---|
| *Prawn* | | | |
| Mean final weight (g) | $19.20 \pm 0.75$ | $19.00 \pm 0.58$ | 0.733 |
| Survival (%) | $30.00 \pm 0.90$ | $17.00 \pm 0.63$ | 0.000 * |
| Final biomass (Kg) | $0.580 \pm 0.040$ | $0.323 \pm 0.025$ | 0.001* |
| *Tilapia* | | | |
| Mean final weight (g) | $170.32 \pm 0.17$ | $172.10 \pm 1.55$ | 0.120 |
| Survival (%) | 100 | 100 | 1 |
| Final biomass (Kg) | $17.033 \pm 0.015$ | $17.213 \pm 0.155$ | 0.116 |
| *Lettuce* | | | |
| Final Mass (kg) | $1.90 \pm 0.37$ | $1.65 \pm 0.42$ | 0.482 |
| *Total* | | | |
| Final biomass (Kg) | $19.51 \pm 0.35$ | $19.18 \pm 0.26$ | 0.256 |

Data presented as mean $\pm$ standard deviation (n = 3). Asterisks (*) indicate statistical differences ($p < 0.05$).

### 3.3. Chlorophyll Compounds in Lettuce (Lactuca sativa L.)

The results of these compounds in lettuce (*Lactuca sativa* L.) are shown in Table 3. Chlorophyll-a and chlorophyll-b were higher after treatment with additional nutrition than without nutritional addition for the soilless culture. However, no significant differences were found in the statistical analysis for both treatments.

**Table 3.** Chlorophyll a and chlorophyll b of the culture in the aquaponic system over eight weeks.

| Compound | NN | MN | *p*-Value |
|---|---|---|---|
| Chlorophyll a | $23.29 \pm 1.88$ | $25.98 \pm 5.06$ | 0.436 |
| Chlorophyll b | $10.495 \pm 0.135$ | $10.510 \pm 2.140$ | 0.991 |
| Total | $33.785 \pm 2.015$ | $36.495 \pm 7.205$ | 0.564 |

Data presented as mean $\pm$ standard deviation (n = 3).

### 3.4. Bacterial Community

The presence of bacteria that can be cultured in each condition was determined depending on the diversity of the phyla obtained with the OTU analysis. Proteobacteria (24.96%) followed by Bacteroidetes (20.89%) was the order of those with the largest relative abundance in the aquaponic system without the addition of nutrients (NN). In comparison, Proteobacteria (27.69%) followed by Bacteroidetes (20.14%) was the order of the largest relative abundance in the aquaponic system with the addition of nutrients (MN) (Figure 3). Using the OTU analysis, 21 genera were found for the genus in both conditions. Flavobacterium (6%), Cetobacterium (7%), and Aurantimicrobium (2%) accounted for the most significant proportion in the aquaponic system without the addition of nutrients (NN). In comparison, Flavobacterium (13%) followed by Cetobacterium (6%) and Aurantimicrobium

(6%) was the order of those with the most significant relative abundance in the aquaponic system with the addition of nutrients (MN) (Figure 4). Table 4 shows the presence or absence of culturable microorganisms under both conditions.

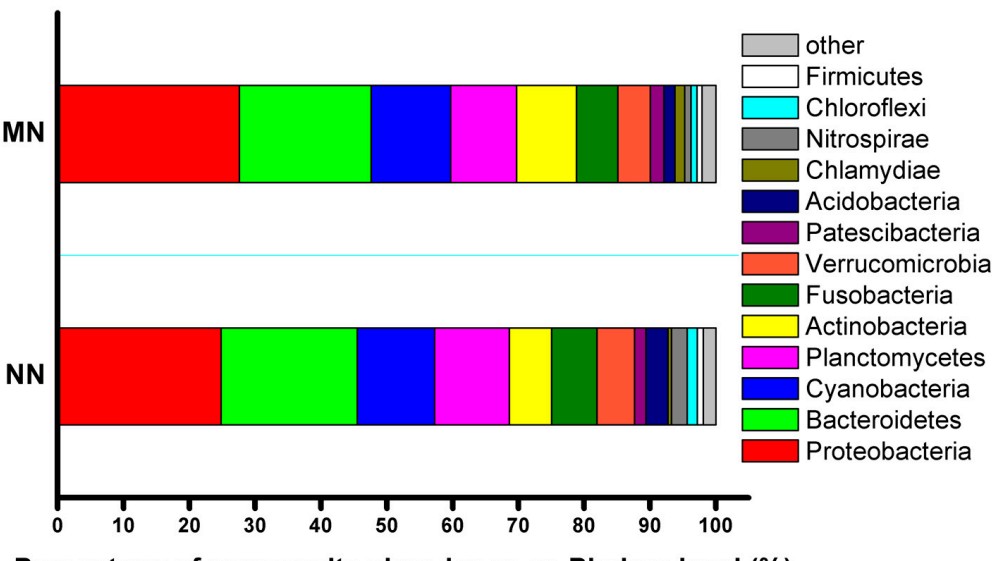

**Figure 3.** Microbial composition during the experiment. OTU analysis based on the phylum level (n = 3).

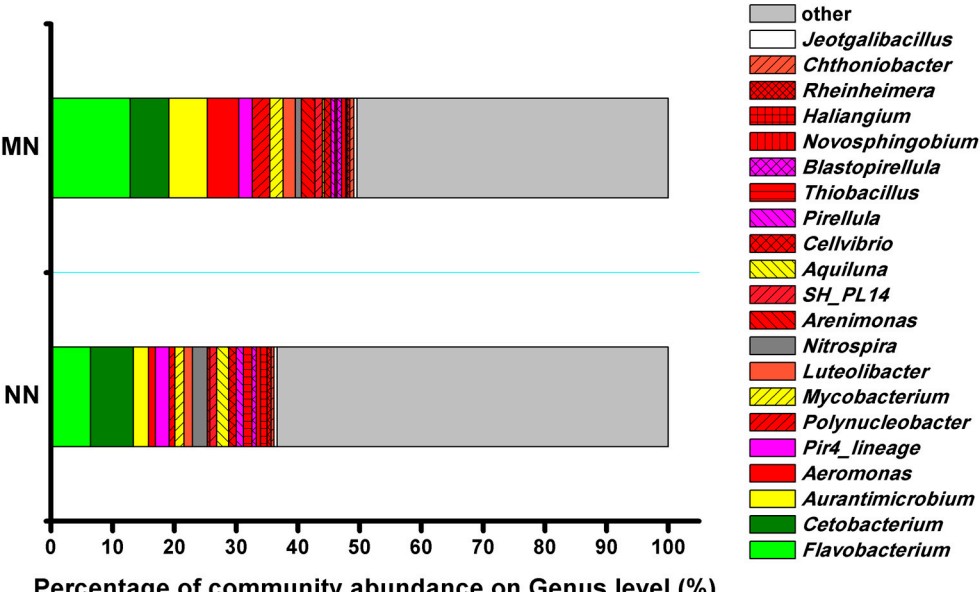

**Figure 4.** Microbial composition during the experiment. OTU analysis based on the genus level. The same color on the genus level indicates from same phylum (n = 3).

**Table 4.** Primary bacterial composition in the experimental aquaponic system. OTU analysis based on the genus level.

| Bacteria | NN (%) | MN (%) |
|---|---|---|
| Nitrogen cycle-related bacteria | | |
| *Aurantimicrobium* [23] | 2.45 | 6.22 |
| *Pir4_lineage* [24] | 2.25 | 2.20 |
| *Nitrospira* [25] | 2.40 | 0.96 |
| *Arenimonas* [26] | 0.43 | 2.19 |
| *SH_PL14* [4] | 1.10 | 1.22 |
| *Aquiluna* [27] | 1.95 | 0.33 |
| *Cellvibrio* [28] | 1.18 | 1.07 |
| *Pirellula* [29] | 1.20 | 0.67 |
| *Thiobacillus* [30] | 1.35 | 0.30 |
| *Blastopirellula* [31] | 0.71 | 0.75 |
| *Novosphingobium* [32] | 0.71 | 0.65 |
| *Haliangium* [33] | 1.07 | 0.20 |
| *Chthoniobacter* [34] | 0.44 | 0.63 |
| Probiotics | | |
| *Cetobacterium* [35,36] | 6.89 | 6.26 |
| *Polynucleobacter* [37] | 0.94 | 2.83 |
| *Luteolibacter* [38] | 1.39 | 2.02 |
| *Rheinheimera* [39,40] | 0.67 | 0.49 |
| *Jeotgalibacillus* [41] | 0.53 | 0.51 |
| Pathogen | | |
| *Flavobacterium* [42–44] | 6.45 | 12.87 |
| *Aeromonas* [45] | 1.12 | 5.09 |
| *Mycobacterium* [46,47] | 1.44 | 2.12 |

A total of 170 genera were identified in both experiments, and only 21 genera were mentioned in the table (>0.5% of the population).

## 4. Discussion

In an aquaponic system, water quality is a limiting element. Maintaining appropriate water quality will ensure that the performance of the animals and plants in the system follows our goal. This seems to be difficult in a connected system since the quality of the water moving in the hydroponic and aquaculture subsystems is the same and continuously flowing in both systems. One main water quality parameter that is challenging to control at optimal values is pH. The optimal pH for plants will be different from that for fish, in general. The plant pH should be 5.5, and the fish pH should be around 7.5 [48]. In this case, for soilless lettuce culture, the pH is around 6.5–7.0 [16]. Our results show that a pH of around 7.5 should be optimal for fish but less for lettuce soilless culture. The survival rate of 100% for tilapia (Table 2) is acceptable for fish living in these aquaponic systems. Tilapia is a widely used fish in aquaponic systems due to its ability to adapt to a range of water conditions and its hardiness [49]. These fish must be able to tolerate the high levels of nutrients in the water, which are added to promote plant growth [50]. On the other hand, the survival rate for giant freshwater prawns (Table 2) is 30.00 ± 0.90% for the non-treatment (NN) and 17.00 ± 0.63% for the with-treatment (MN). Both of these results are not as good as those found in another study of around 55% [51], thus survival should be affected by another factor.

Another significantly different water quality between both treatments is phosphate ($PO_4^{3-}$) (Table 1). Phosphorus in aquaculture water bodies is often in an un-proportional amount [52]. Considering aquaculture pond contents are frequently more phosphorus-rich than natural waters, and because their discharge could contaminate receiving waters with phosphorus compounds and promote increased growth of plants, phosphorus is essential [53]. Around 30–65% of the phosphorus in the fish diet is still inaccessible to plants because it is fixed in the solid excretions, which are subsequently filtered out

mechanically [54]. On the other hand, plants cannot easily access the kinds of phosphate found in fish waste. Because it is the only form that plants can absorb and assimilate, inorganic ionic is needed in adequate levels [55]. Above pH 7.0, inorganic phosphate bonds to calcium; hence, aquaponic systems should be careful about keeping pH levels near 7.0. Various insoluble derivatives of calcium phosphate can accumulate in sediment as pH levels rise above 7.0 [55,56]. The amount of phosphorus delivered by fish meal might vary regarding to the production state in a system that grows lettuce. Based on how the system is set up, roughly 100% of the phosphorus water can be recycled in the plant biomass [57]. In this study, there was much less in the treatment with added nutrition than in the treatment without nutrition (Table 2). It appears that phosphorus tends to be restricting and might thus inhibit plant growth including lettuce production [13,58]. P has a few distinct qualities; one is its poor availability brought on by slow absorption and increased soil fixing. This all indicates that P may pose a significant growth-limiting challenge to plants. Although using animal waste and chemical P for fertilization on cropland has increased crop yield and inorganic P richness, they have also damaged the ecosystem throughout the years [59].

Although feed remains the primary source of phosphorus for prawn ponds [60], the higher phosphate concentration in this study (Table 1) is likely influenced by the phosphate levels in the water flow. The added nutrition contains 90-g Monopotassium Phosphate ($KH_2PO_4$) and 37-g Monoammonium Phosphate ($NH_4H_2PO_4$) in every 1000 L; this should be one of the sources of phosphate in the water aquaponic system with additional nutrition. When phosphate in water increases, it will positively affect AOB communities [61,62]. Trends in the significance of high phosphate and AOB (Table 1) clearly define that AOB in an aquaponic system with supplemental nutrition has a more significant population than without supplemental nutrition. As soon as particulate or soluble organic phosphorus breaks down into dissolved inorganic phosphorus and sinks to the bottom, substrate algae and phytoplankton take up the phosphorus [63]. Following phosphate dosing, a shift in the microbial community towards organisms that can utilize or metabolize phosphate was observed, including an increase in certain types of bacteria. [64].

A total of 13 phyla and 21 genera of known bacteria in this non-phosphorus-adding treatment system (>0.5% from population abundance) is dominated by N-related bacteria, which represent the majority of all bacteria in systems, followed by probiotic bacteria and pathogens (Figure 5). A total of 11 genera of bacteria involved in nitrogen cycles have been found (Table 4). This type of bacteria has the potential to successfully co-culture plant and animal symbiosis. Their primary function is converting the ammonia from the excrement and uneaten fish/prawn diet to nitrate and nitrite, allowing aquatic plants to use the end material more easily for growth. One species known to obtain energy through the anaerobic oxidation of ammonia is one of the *Pirellula*-like organisms considered typical root-associated bacteria [24,65]. They act as catalysts for significant changes in the global carbon and nitrogen cycles. *Pirellula* sp. strain 1 contains a gene that encodes a bacterial hemoglobin that is thought to detoxify nitric oxide (NO) by oxidation. Some carbon-starvation proteins, such as DNA-protection proteins, have homologs in the genome [29]. *Nitrospira* would fit into the comammox's theoretically projected ecological niche. Biofilm or floc production is a feature of the engineered systems examined in this work. In biofilms, diffusion barriers and ammonium or nitrite concentration gradients could form niches with limited substrate influx, allowing comammox to outcompete incomplete nitrifiers. Diverse nitrifier communities may be supported by complex biofilm or floc designs with multiple microenvironments [25].

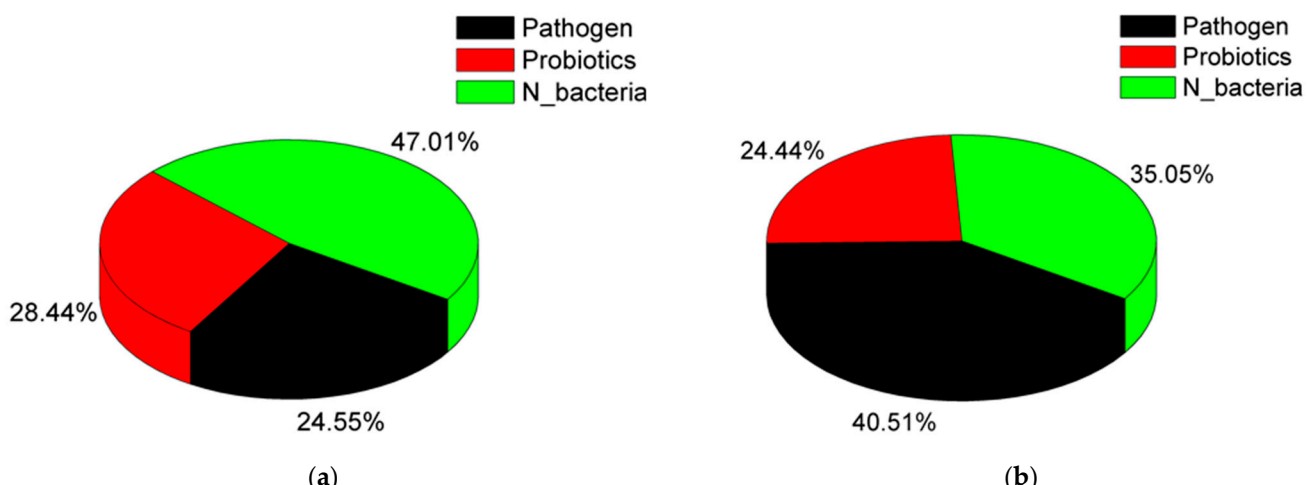

**Figure 5.** The average percentage is known bacteria cultivable in aquaponic systems for eight weeks (**a**) without treatment and (**b**) with additional nutrition. (n = 3).

Next, the phosphorus-adding nutrition treatment group is dominated by pathogens, which represent the majority of all bacteria in systems, followed by N-related and probiotic bacteria (Figure 4). A total of three genera of pathogen bacteria have been found (Table 4). Even though just three genera have been found, they are likely to impact fish and prawn culture significantly. The total density of harmful bacteria that developed in the aquaponic system with nutrients added (MN) was substantially more significant than that of harmful bacteria that developed in the aquaponic system without nutrients added (NN). *Flavobacterium*, the most significant percentage in this group, could be one of the factors behind the dramatically varied survival rates of freshwater prawns in this study. In several studies, *Flavobacterium*, known as a pathogen [43,44], can be found in the early embryonic phase of *M. rosenbergii* when in the blastula stage [66]. According to some studies, *Aeromonas* is one of the diseases bvastria that has put the huge freshwater prawn growth system at risk [67–69]. The damage produced by mycobacteria invasion and spread had not yet manifested clinical indications, such as aberrant conduct. The cytoplasm of fixed phagocytes of this animal did not contain acid-fast bacteria. According to the histopathology findings, *M. rosenbergii's* predominant host response to mycobacteria infection was hemocyte encapsulation rather than phagocytes [46].

Chlorophyll, the principal pigment of leafy green vegetables, is essential in determining their health condition. Because most nitrogen is absorbed in leaf chlorophyll, the nutritional status of leafy green vegetables can also be assessed by quantifying chlorophyll content. In this study, the amount of chlorophyll extracted using both procedures (grinding–settling and immersion) had no significant difference between the aquaponic systems with and without added nutrition (Table 3), which is consistent with earlier studies [70,71]. It has been proven that chlorophyll activity is still regular.

## 5. Conclusions

Adding a fertilizer containing phosphate directly into a body of water can significantly increase phosphorus levels in an aquaponic system. This strongly supports the observation that AOB and NOB alter their community. An increase in pathogenic bacteria, particularly Flavobacterium and Aeromonas, was observed as a result of nutrient addition. However, there is no significant change in water quality (DO, pH, temperature, ammonia, nitrate, nitrite) or the growth performance of tilapia and lettuce. The only result not in favor of this system is the giant prawns' meager survival rate, which could have been affected by pathogenic bacteria in the nutrient-rich system. Both *Flavobacterium* and *Aeromonas* are known pathogens in most shrimp or prawn ponds. Further research is needed on the sources of phosphate in aquaponic systems that need to be controlled for their forms that are rather accessible for plants, but that should not support the proliferation of microbial

pathogens. Additionally, using other species that have a high tolerance to fluctuations in macro- and micro-nutrients, such as catfish, may be considered during the experiment.

**Author Contributions:** Conceived the idea and developed the theory and experimental design, M.-C.H. and B.C.; measurements, planning and supervised the work, M.K.; processed the experimental data, performed the analysis, drafted the manuscript, and designed the figures, S.N. and M.K.; manufactured the samples and characterized them with statistics, S.N. and T.-Y.S.; co-aided in interpreting the results and worked on the manuscript, M.-C.H. and T.-M.W. All authors have read and agreed to the published version of the manuscript.

**Funding:** This work was supported by the Ministry of Science and Technology, Taiwan (MOST 107-2311-B-992-003-MY3) and the MOE Teaching Practice Research Program (PSR1100590). The funders had no role in the study design, data collection and analysis, decision to publish, or preparation of the manuscript.

**Institutional Review Board Statement:** This animal study was reviewed and approved by the Institutional Animal Care and Use Committee of National Pingtung University of Science and Technology (approval code NPUST-108-017, approved on 17 March 2019).

**Data Availability Statement:** The data presented in this study are available on request from the corresponding author.

**Acknowledgments:** We acknowledge the T-M, Wu's lab. in Taiwan for kindly providing of the source of tilapias and also for technical and equipment support as well as for providing comments that greatly improved the manuscript.

**Conflicts of Interest:** The authors declare no conflict of interest.

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
