# Peer review of "Effects of Phosphate-Enriched Nutrient in the Polyculture of Nile Tilapia and Freshwater Prawn in an Aquaponic System"

_fishes, doi:10.3390/fishes8020081_

Round 1
Reviewer 1 Report
The study aimed to determine if supplementation with an exogenous source of phosphorus could have a possitive impact in total production in an aquaponic system. The combination of organisms selected by the authors was formed by tilapila, giant prawn and lettuce.
There are important defficiencies in the description of the methodology that precludes publication of the study under its present form:
M&M section:
The two first paragraphs describing the experimental setup are repetitive. Please, resume and condense. In fact, the general description of the experimental arrangement is not clear. I suggest the authors rearrange the sentences and:
- First describe the general structure of the aquaponics system
- Secondly describe the type of treatments to develop. Authors should indicate clearly the process followed for enrichment; which compound did they use? At what dosage? Did they make only an initial enrichment or they repeated phosphorus supply along the experiment? Did they make any regular test to ensure that a constant concentration was maintained?
- Thirdly, indicate in detail how many replicates used per treatment and how many individuals of each species (fish, prawns and lettuce) were used in each replicate. Also indicate the total time for the experiment explaining the duration of the conditioning period and the experiment itself.
- Finally, indicate the sampling pattern used to collect the values of water quality along the experiment. In this sense, if the main factor considered in the experiment is the extra supply of phosphorus, this should be monitored more frequently (i.e. on a daily basis, at least during the initial stages of the experiment). In fact, a time plot showing variations in this parameter should be included
Regarding microbiology, authors should indicate at what moment of the experiment were the samples collected and how many samples were taken for each treatment. Also detail number of samples and processing of such samples used for determination of chlorophyll. Authors indicate the use of “three extracts”; what does this mean?
RESULTS:
Fig 3. Indicate % numbers in the X axis
Discussion > Authors focus the discussion in the negative effect of phosphorus supplementation on the microbiota of the system that increases the level of pathogenic bacteria. They should also explain why no effect was observed in tilapia
They should also develop a more detailed discussion considering the effect of the type and dosage of P used in their experiment (not indicated) as well as the possibility of having obtained a different response when using a different combination of organisms.
Author Response
Please see the attachment, thanks.

Reviewer 2 Report
In this study, the authors investigated the effects of phosphate-enriched nutrient in the polyculture of Nile tilapia and freshwater prawn in an aquaponics system. They showed that adding phosphate increases phosphorus levels in an aquaponics system, and changes microbial community and species growth performance. I think that the results presented here are beneficial to establish an effective aquaponics system. My comments are given below.
Line 99: Correct “0,5” to “0.5”.
Lines 124–127: The authors should explain “Water quality monitoring” in more detail (e.g., collection of water samples, measurement principle of ammonia, nitrite, nitrate and phosphate, and QA/QC). As mentioned in the Introduction section, calcium may control the phosphate concentration in the system water. Why did not the authors measure the calcium concentration?
Table 1: The significant figures for given values are too many. They are at most two or three due to large variations.
Line 185: Eliminate “(Figure 2)”
Line 225: Correct “30.00 + 0.90%” to “30.00 ± 0.90%”.
Line 256: This sentence has a problem with English.
Lines 259–261: I do not understand the meaning of this sentence.
Lines 261–263: This sentence has a problem with English.
Line 312 Eliminate “and” after “NOB”.
Lines 312–313: This sentence has a problem with English.
Author Response
Please see the attachment, thanks.

Round 2
Reviewer 1 Report
Authors have addressed most of the comments. The paper could be suitable for publication